# A model-based evaluation of the efficacy of COVID-19 social distancing, testing and hospital triage policies

**Audrey McCombs**[1], **Claus Kadelka**[2]*

**1** Department of Statistics, Iowa State University, Ames, IA, United States, **2** Department of Mathematics, Iowa State University, Ames, IA, United States

* ckadelka@iastate.edu

**Data Availability Statement:** The complete Python implementation of the model is available at Github at https://github.com/ckadelka/COVID19-network-model.

## Abstract

A stochastic compartmental network model of SARS-CoV-2 spread explores the simultaneous effects of policy choices in three domains: social distancing, hospital triaging, and testing. Considering policy domains together provides insight into how different policy decisions interact. The model incorporates important characteristics of COVID-19, the disease caused by SARS-CoV-2, such as heterogeneous risk factors and asymptomatic transmission, and enables a reliable qualitative comparison of policy choices despite the current uncertainty in key virus and disease parameters. Results suggest possible refinements to current policies, including emphasizing the need to reduce random encounters more than personal contacts, and testing low-risk symptomatic individuals before high-risk symptomatic individuals. The strength of social distancing of symptomatic individuals affects the degree to which asymptomatic cases drive the epidemic as well as the level of population-wide contact reduction needed to keep hospitals below capacity. The relative importance of testing and triaging also depends on the overall level of social distancing.

## Author summary

Public health policies implemented to reduce the effects of COVID-19 can interact with each other, enhancing or undermining the effects of other policies employed simultaneously. Here, we present a mathematical model that incorporates many of the important characteristics of the outbreak, including differences in risk behavior and social activity due to demographics, and uncertainties related to asymptomatic cases. Our results suggest that reducing random community encounters is more important than reducing personal contacts, and that testing low-risk versus high-risk symptomatic individuals is most effective. Results also suggest that the effectiveness of a particular policy choice depends on what other policies are concurrently employed, and that policy makers should account for these interactions when considering which guidelines to implement.

**Funding:** The author(s) received no specific funding for this work.

**Competing interests:** The authors have declared that no competing interests exist.

## Introduction

In response to the current COVID-19 pandemic caused by SARS-CoV-2, public health organizations have deployed plans previously developed in anticipation of a major influenza outbreak [1, 2]. These plans describe, among other things, policies related to non-pharmaceutical interventions such as social distancing and the allocation of scarce healthcare resources. Policy guidelines for testing individuals for exposure to SARS-CoV-2 are less well-developed, and a shortage of tests has hampered response efforts in many countries.

Our model explores the simultaneous effects of policy choices in three domains: social distancing, testing, and hospital triaging. While many models examine the effects of a single policy domain on the dynamics of an infectious disease, e.g., [3–6]; studies that examine multiple policy domains together exist but are not as common [7–10]. Considering policy domains together can provide crucial insight into how different policy decisions interact. Policies from different domains may enhance or undermine each other when implemented together. Our model provides a tool for investigating these interaction effects.

The spread of an infectious disease can be strongly influenced by social behavior [3, 5]. Because the effects of COVID-19 seem dependent on demographics such as age, allowing for differences in social interactions due to demography and perceived risk is an important aspect of modeling this disease. Triage decisions also affect disease dynamics after hospitals surpass their care capacity. Possible strategies for determining who will receive scarce healthcare resources such as ventilators include: 1) first-come first-served, 2) randomized allocation (e.g., lottery), and 3) clinical judgment [1, 2, 11]. Testing guidelines from the Centers for Disease Control and Prevention (CDC) have evolved rapidly over the course of the pandemic [12]. For viral testing, CDC guidelines currently prioritize hospitalized patients, healthcare workers, first responders, and people in congregate living settings such as long-term care facilities and prisons, if those individuals show symptoms.

Classical compartmental differential equation models (e.g., SIR and SEIR models) are an invaluable tool for understanding the general course of an infectious disease at the population level [13]. However, these models assume homogeneous mixing of the population as well as constant transition rates [14]. Neither assumption is valid for real interaction networks and COVID-19 [15, 16], and can result in significantly different disease dynamics [17, 18]. Our study avoids both these pitfalls by implementing a stochastic compartmental disease model evaluated on a network, yielding more realistic disease dynamics [19].

We simulated how SARS-CoV-2 spreads through an abstract community of 10,000 individuals, where physical interactions with private social contacts (family, friends, coworkers, etc.) and public, random encounters (shopping, banking, etc.) were represented as small-world and fully-connected networks, respectively (Fig 1A). Upon infection, susceptible individuals (S) transition through contagious compartments of the model (exposed (E), asymptomatic (A), symptomatic (I) and hospitalized (H)), finally resulting in death (D) or recovery (R) (Fig 1B). The model incorporates important characteristics of the current COVID-19 outbreak such as asymptomatic cases and early and asymptomatic transmission of the virus (Fig 1C). We further modeled the differential risk associated with COVID-19 by distinguishing between high-risk individuals (older individuals or individuals with known comorbidities [20]) and low-risk individuals (younger individuals without known comorbidities). Behavioral differences associated with risk level and infection status are captured by the model, as well as the reduction in care caused by hospitals operating beyond their capacity (Fig 1D).

The primary goal of this study was to evaluate the possibly interacting effects of various policies regarding social distancing, triaging, and testing on reducing COVID-19-related mortality. The abstract nature of our model enables a reliable evaluation of the relative qualitative

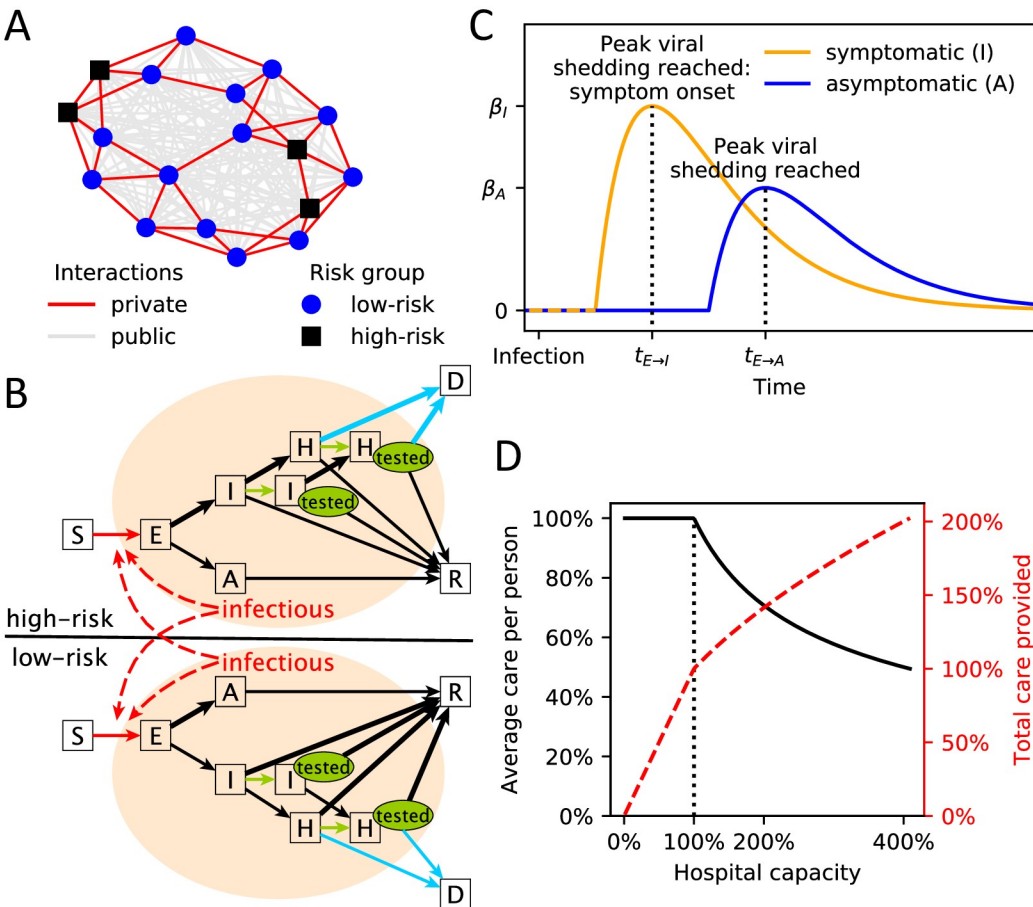

**Fig 1. Graphical model summary.** (A) Example of the two-layer interaction network used in this study. The private, small-world network (red edges) is shown on top of the public, fully-connected network (gray edges). Low-risk (blue circles) and high-risk (black squares) individuals are distinguished. (B) Illustration of the stochastic transmission model with compartments S = susceptible, E = exposed, A = asymptomatic, I = symptomatic, H = hospitalized, R = recovered, D = deceased. Individuals in I and H may receive a positive test (green "tested" oval). Edges that are influenced by policy decisions are colored: red = social distancing, green = testing, blue = hospital triage. Branching probabilities at E, I and H are risk-group dependent and the edge of the respectively more likely transition is thicker. (C) Illustration of the time-dependent transmission rate of an exposed individual increasing until peak viral shedding, which coincides with transition to compartment I (if symptomatic) or A (otherwise). (D) Average care per person (blue solid line) and total care provided (red dashed line) by a health care system with a capacity threshold of 100% operating at a certain level of (over)capacity. Once the capacity threshold is reached, the average care per person is $1/\sqrt{\text{hospital capacity}}$.

efficacy of different policy decisions, despite current uncertainty in key virus and disease parameters [21]. The model can be easily updated and expanded once more accurate parameter estimates are available, and can be tailored to a specific community or country in order to evaluate the effects of policies being considered for implementation.

## Results

The epidemiological outcome measures from our model fall within the range of current estimates [3, 4, 22]: an average initial basic reproductive number ($R_0$) of 2.76 and an average disease generation time of 5.29 days. Higher transmissibility and higher $R_0$ values were associated with shorter generation times, which in turn were associated with hospital overcapacity and a faster spread of the virus (S1 Fig). All model outcomes we investigated (S2 and S3 Figs) were

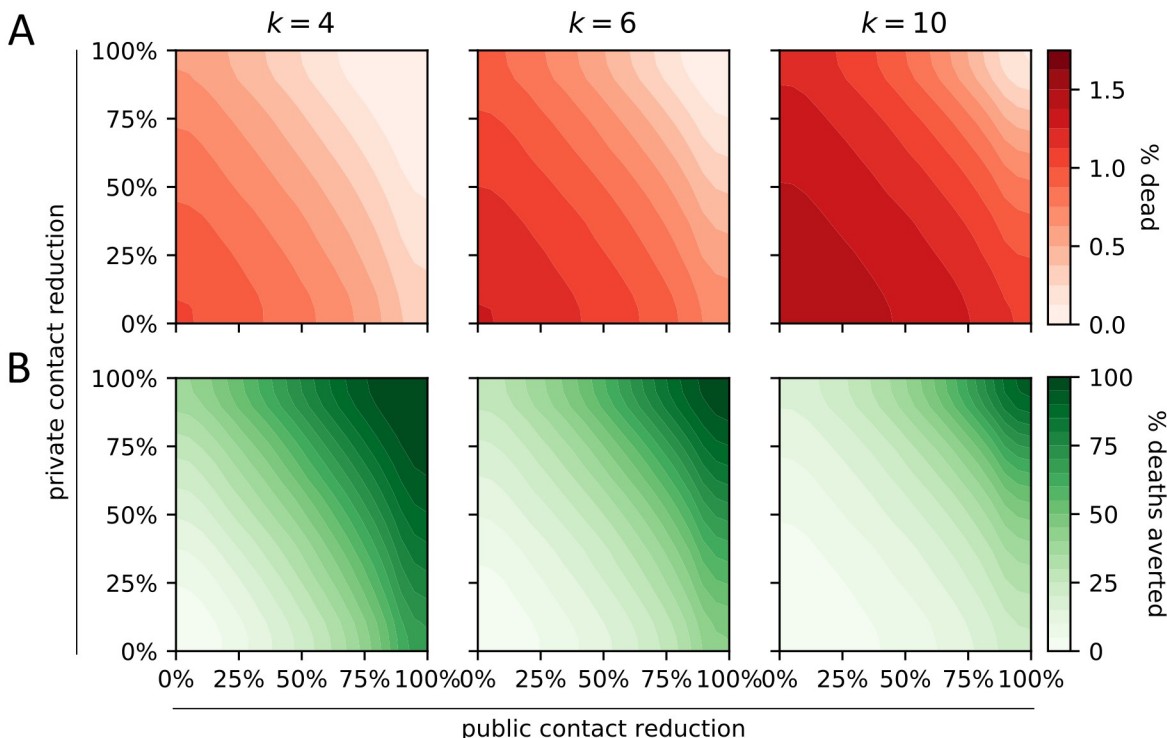

**Fig 2. Impact of social distancing on COVID-19-related mortality.** Effect of population-wide private and public contact reduction on the average proportion dead (A, red) and the average percentage of averted deaths (B, green). The latter is computed by comparison with no contact reduction. Results are shown for communities with *k* private and *k* public average contacts per day, for different values of *k*.

highly correlated (S4 Fig), therefore in reporting our results we focus mainly on COVID-19-related mortality.

As expected, social distancing measures reduced the number of deaths. We considered interaction networks with the same number of public and private contacts; a reduction in public contacts had a stronger effect on the number of deaths than an equal reduction in private contacts (Fig 2A). Furthermore, a given reduction in private and public contacts more successfully reduced the number of deaths in less-connected networks (Fig 2B). Asymptomatic cases (truly asymptomatic or not yet symptomatic) caused the most infections, which explains the ease with which SARS-CoV-19 is spreading across the world (Fig 3). The proportion of infections caused by asymptomatic cases was influenced more by the behavior of symptomatic individuals than by the actual rate of asymptomatic cases. That is, a given increase in contact reduction of symptomatic individuals caused a bigger change in the proportion of infections caused by asymptomatic cases than an equal increase in the proportion of asymptomatic cases. The behavior of symptomatic individuals also affected the level of population-wide contact reduction needed to keep hospitals operating within capacity (Fig 4A). Even assuming perfect isolation of symptomatic individuals, however, hospitals quickly surpassed their capacity unless very strong levels of population-wide social distancing were implemented.

We compared different triage policies found in the literature [11, 23, 24] to a worst-case scenario in which hospitals operating at overcapacity provided an imperfect but equal level of care to all patients (Fig 4B). Current hospital policy of prioritizing the care of the least-severely infected patients (based on clinical judgment) proved the most successful in reducing COVID-19-related mortality. Filling empty beds on a first-come first-served basis was less successful,

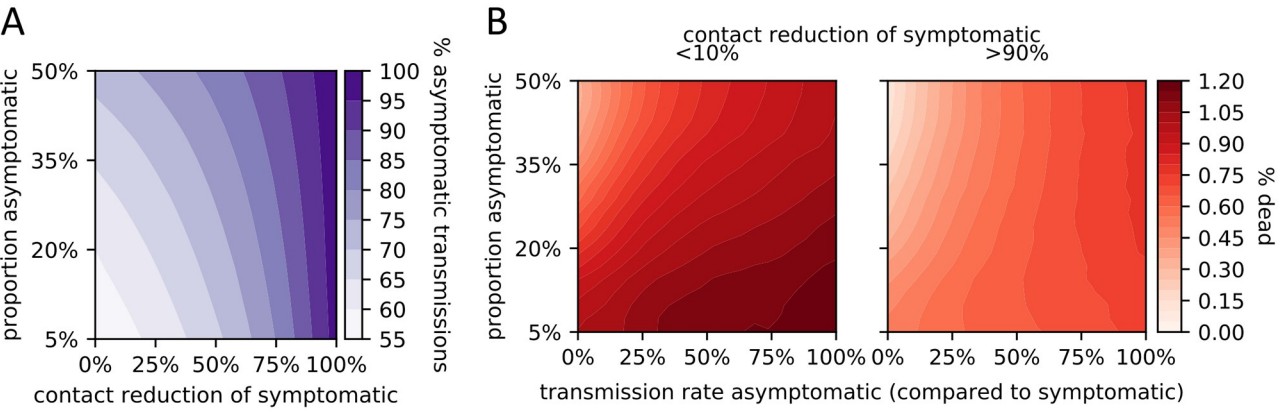

**Fig 3. Impact of asymptomatic cases.** (A) Effect of the proportion of asymptomatic cases (y-axis) and the contact reduction by symptomatic cases (x-axis) on the percentage of infections caused by asymptomatic cases. (B) Effect of two unknowns on the proportion dead: the proportion of asymptomatic cases (y-axis), and their relative transmission rate (compared to that of symptomatic; x-axis). Two scenarios are considered: low ($<$ 10%; left panel) and high ($>$ 90%; right panel) contact reduction by symptomatic individuals.

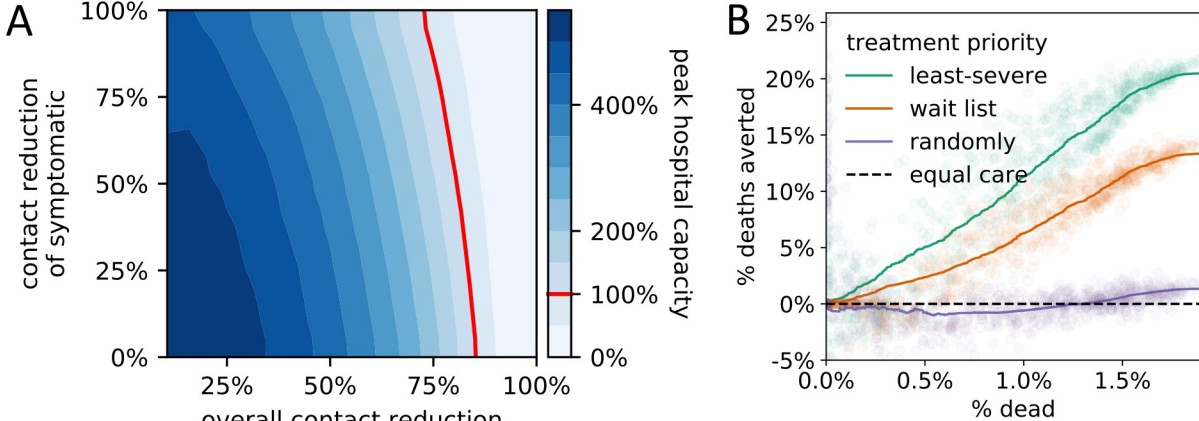

**Fig 4. Peak hospital capacity and the effect of triaging policies.** (A) Effect of population-wide contact reduction and additional contact reduction of symptomatic individuals on the projected peak hospital capacity. To the left of the hospital capacity threshold (red line), not all hospitalized individuals receive perfect care, resulting in longer recovery times and higher mortality. (B) Relative efficacy of four hospital triage policies at varying degrees of outbreak severity summarized by the average proportion of deaths (x-axis). Efficacies are computed by pairwise comparison of the projected death count with the imperfect-but-equal-care scenario.

while filling empty beds randomly resulted in a very similar, high number of deaths as an imperfect-but-equal-care scenario.

Increasing the availability of testing and reducing the delay between test administration and results both reduced the total number of deaths (Fig 5A). Positively tested individuals self-quarantine; the effectiveness of self-quarantine (i.e., level of contact reduction) had the strongest effect on the death count under an efficient testing regime (Fig 5B). Our model only considered the effect of testing on social behavior and not clinical outcomes, but within this framework we found that prioritizing testing of low-risk individuals consistently reduced the number of deaths more than testing high-risk individuals first (Fig 5C). Furthermore, within each risk-group, testing recently-infected individuals first was more effective than prioritizing individuals who have been infected longer.

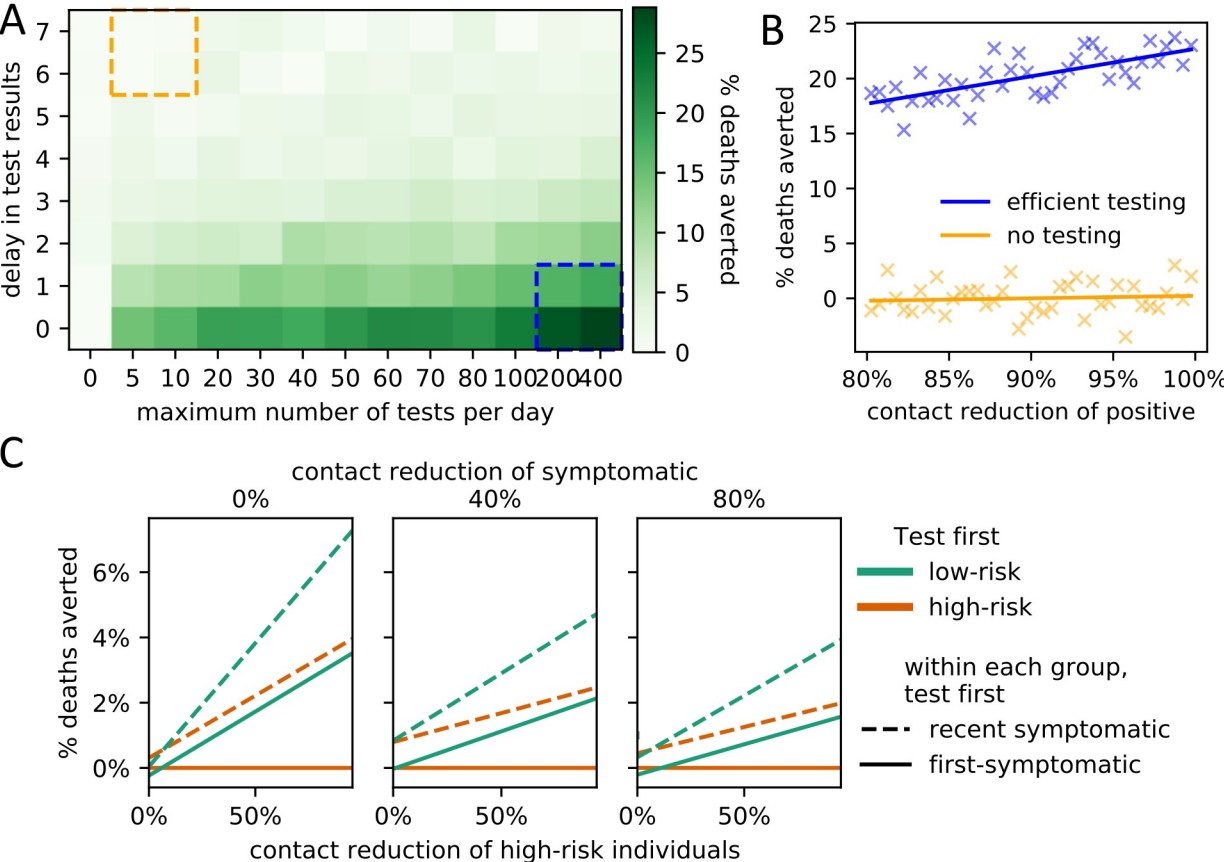

**Fig 5. Impact of testing policies on COVID-19-related mortality.** (A) Effect of increased numbers of tests (x-axis) and processing delays (y-axis) on the average percentage of averted deaths (compared to no testing; left-most column). (B) Effect (linear regression) of the average contact reduction by individuals with positive test results on the percentage of averted deaths under an efficient testing scenario (blue box lower right in A) and an inefficient testing scenario (orange box upper left in A). (C) Impact of policies regarding testing prioritization of symptomatic individuals on the average percentage of averted deaths (compared to the worst policy). The primary policy decision involves which risk group to prioritize (low-risk (green) or high-risk (orange)). The secondary policy decision involves who to test first within each risk group (newly symptomatic (solid lines) or first-symptomatic (dashed lines)). The results are stratified for three different levels of additional contact reduction due to symptoms (subplots) as well as for varying levels of additional contact reduction of high-risk individuals (x-axis). Linear regression fits are shown.

The efficacy of a particular policy choice depended on what other policies were implemented (Fig 6). When social distancing was low, the choice of triage policy affected the number of deaths more than the level of testing. When social distancing was high however, triaging choices hardly mattered, while testing became proportionately more important.

## Discussion

Consistent with other COVID-19-related modeling studies (e.g. [3, 4]), our findings generally support current policies to reduce the public health impacts of SARS-CoV-2 and COVID-19 infection.

Our comparison of interaction networks suggests a possible refinement of current social distancing policies. We found that for equally connected networks, a reduction in public (random) contacts had a stronger effect on the number of deaths than an equal reduction in private contacts (Fig 2A). This may be because public encounters allow the virus to spread to otherwise disparate parts of the social network, while private contacts enable only local spread.

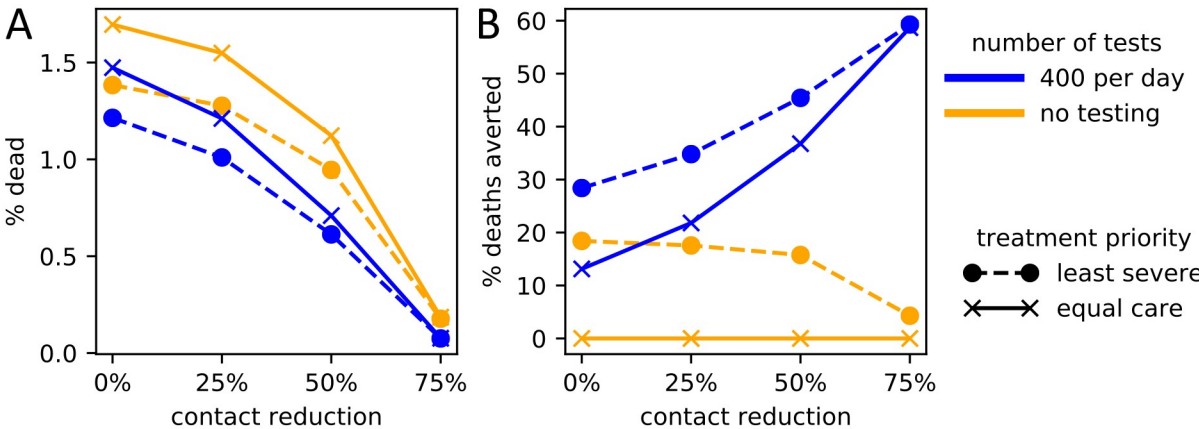

**Fig 6. Interactions of the three policy domains on COVID-19-related deaths.** The level of social contact reduction (x-axis) is plotted against combinations of testing policies (solid lines, gold: maximum testing, blue: no testing) and triage policies (dashed lines, circles: treat least severely infected first; solid lines, crosses: imperfect but equal care). Y-values are averages across 1,000 parameter choices with 100 replicates each. (A) Policy choices versus proportion dead (y-axis). (B) Relative effects of policy choices in reducing the proportion dead (y-axis), compared to the worst-case scenario of no testing and imperfect-but-equal-care triage policy.

Random encounters thereby prevent dead ends for the disease. Our findings agree with previous results suggesting that strong clustering in a private network reduces the potential for transmission at the population scale [25, 26]. Based on these findings, future public health notices could emphasize the need to reduce contacts specifically with random individuals.

A comparison of interaction networks with different levels of social connectivity may provide a partial explanation for observed country-specific differences in the impact of the virus (Fig 2B). A seminal study found strong differences in the average number of daily interactions in different countries in Europe, where Italy's average per-day contact rate was 20 while Germany's was only 8 [27]. Our results suggest that, all other things being equal, more social societies such as Italy's must reduce their contacts more in order to attain the same reduction in deaths. A potentially fruitful avenue for future research might investigate whether these results hold more broadly across different cultures and different sectors of society.

CDC guidelines for testing have evolved rapidly over the course of the pandemic, changing several times between February and June, 2020 [12]. We model only two of the many factors that could be considered when developing testing policy—the risk of developing severe symptoms, and the length of time since symptom onset. Our model does not account for potential improvements in clinical outcomes due to testing; a positive test only affects behavior through a reduction in social activity. Under this framework, prioritizing the testing of low-risk individuals more effectively reduces the overall death count than prioritizing the testing of high-risk individuals. Low-risk individuals have higher average contact rates [27] (in our model, they practice less social distancing than high-risk individuals), so reducing the social activity of infected low-risk individuals (due to positive test results) slows the spread of the disease. Recognizing these differences in social activity levels may be important to the CDC when considering testing guidelines.

An analysis of policy interactions further highlights the importance of testing. Several results suggest that the efficacy of a particular policy choice depends on what other policies are concurrently implemented. The strength of social distancing of symptomatic individuals affects the degree to which asymptomatic cases drive the epidemic (Fig 3), as well as the level of population-wide contact reduction needed to keep hospitals below capacity (Fig 4A). Examining all three policy domains together, the relative importance of testing and triaging depends

on the level of social distancing (Fig 6). The presence of interactions suggests that quantitative predictions based on the examination of only one policy domain in isolation should be treated with caution. Furthermore, when policy makers consider which guidelines to implement, they should account for other policies that could influence the efficacy of the guidelines under consideration.

To capture important elements of the COVID-19 pandemic, we considered a physical interaction network that differentiates between private contacts and random encounters as well as high-risk and low-risk individuals. Because we were primarily concerned with the interaction of policies in different domains, we did not include many complicating elements of more realistic social networks, such as variability in the number of contacts, variability in clustering motifs, heterogeneity in individual risk behavior (e.g. super-spreaders) [18], and age-associative mixing. Many of these social network characteristics have been shown to influence epidemic spread [7, 19, 25, 26, 28]. For example, the inclusion of super-spreaders in a model makes disease extinction more likely and outbreaks rarer but larger [18]. The final epidemic size has also been shown to increase as clustering decreases and to decrease as the heterogeneity in the number of contacts increases [26]. Additionally, the intensity of physical interactions proved more important in determining the size of an outbreak than the actual number of interactions [25, 28]. While the private small-world network incorporates aspects of clustering in our model, heterogeneity in the number of contacts and the presence of super-spreaders could allow the disease to spread more easily across the network. Inclusion of additional, complicating elements in our network model could change the size of the epidemic and the overall transmission dynamics.

While we modeled the effect of social distancing as changes in interaction probabilities (edge weights in the network), we only considered static networks, which do not evolve over time to capture temporal movement of individuals [25, 29]. Further, we studied a scenario in which the infection begins with a single infected individual and excluded the possibility of ongoing re-introductions. In 36.1% of simulation runs the epidemic thus remained contained ($< 10\%$ of the population became infected). When the epidemic did spread widely, a lack of temporal adjustment of policy choices led to very high numbers of infected individuals, which may explain the observed bimodal distribution of the proportion of infected individuals (S2C Fig). In a more realistic setting, policies such as social distancing efforts are dependent on the state of the epidemic, as is currently observed in countries and states loosening restrictions due to seemingly lower infection numbers.

Many key parameters of the COVID-19 epidemic are still unknown and may vary from community to community. We therefore ran our stochastic transmission model for a large range of parameter settings. Due to the inherent uncertainty in many model parameters however, the absolute values of the model outcomes depend on the particularities of the underlying parameter space. The goal of this study was not to predict the expected absolute number of COVID-19-related deaths; rather, the model is a tool that can be used, despite the uncertainty in key parameters, to compare the efficacy of various policies aimed at reducing mortality. We have focused on relative comparisons of three policy domains, as relative findings are more robust to uncertainty in the underlying parameter space.

In conclusion, the presented model considers a variety of factors important to the current COVID-19 pandemic, including heterogeneous risk factors, varying social distancing behaviors, and the unknown proportion and transmission potential of asymptomatic cases. Importantly, the model can be easily expanded and updated as more details about SARS-CoV-2 and COVID-19 emerge. Our results support current policies to contain the outbreak and suggest possible refinements to public health policy and education. Specifically, because a reduction in public contacts more effectively controls disease spread than an equal reduction in private contacts,

future public health notices could emphasize the need to reduce contacts specifically with random individuals. Our results also provide a possible explanation for why some societies are more successful at containing the outbreak despite the implementation of similar policy measures. Follow-ups to this study could deepen our understanding of how heterogeneity in network structure and risk behavior affect the interplay of policy decisions and disease dynamics.

## Materials and methods

### Physical interaction network

We modeled the spread of SARS-CoV-2 across a multi-layered physical interaction network as an abstract proxy for a town or local community (similar to [30]). We considered two population sizes: N = 1,000 and N = 10,000. Model results were qualitatively identical for both population sizes, therefore we reported only results for a population of 10,000 individuals. We considered a closed population, and given the short time frame of the COVID-19 pandemic also chose not to include birth, deaths, or migration events. We distinguished between two types of interactions: First, private contacts (family, friends, school and work colleagues, etc.) were modeled using a Watts-Strogatz small-world network with average connectivity of $k$ neighbors and 5% probability of edge rewiring (Fig 1A) [31]. Note that the edge rewiring is part of the initial network generation and we did not consider dynamically changing private contact networks. Second, public, random encounters (grocery shopping, banking, etc.) were modeled using a fully connected network [32]. We assumed that, in the absence of an epidemic, an individual has on average the same number of $k$ private and $k$ public interactions, and thus added a weight of $k/(N-1)$ to every edge to represent public interactions. A multinational study found, on average, between 8 and 20 per-person per-day contacts [27], so we considered $k \in \{4, 6 \text{ (baseline)}, 10\}$.

### Modeling COVID-19-specific characteristics

We modeled the differential risk associated with COVID-19 by distinguishing between high-risk individuals (older individuals or individuals with known comorbidities [20]) and low-risk individuals (younger individuals without known comorbidities). Each node represented a high-risk (low-risk) individual with probability $p^{\text{high-risk}} = 1/3$ ($p^{\text{low-risk}} = 2/3$) [33, 34] (see caption of Table 1 for details). Our model includes seven qualitatively different compartments: S = susceptible, E = exposed, A = asymptomatic, I = symptomatic, H = requiring hospitalization due to severe infection, and two final compartments R = recovered and D = died from COVID-19 infection (Fig 1B). The length of time recovered individuals remain immune is currently unknown, however given the short time frame (weeks to months) of this model, we assumed no reinfections. To model the spread of the virus in a fully susceptible population, we initialized the simulation with one random individual in compartment E; all others started in S.

Model inputs related to the specific characteristics of SARS-CoV-2 and COVID-19 were derived from published literature where available (Table 1). Virus and disease parameters without established estimates were included as random variables from broad uniform distributions. We considered time to be discrete with one unit of time corresponding to a day. The length of time individuals spend in contagious compartments was modeled using Poisson random variables with parameters derived from the literature (Table 2). Other distributions frequently used to model wait times, such as the discrete Erlang, log-normal, or the general Gamma distribution, all belong to two-parameter distribution families and thus cannot be estimated from just the mean or median. Furthermore, the probability mass functions of the one-parameter geometric and exponential distribution families peak at zero, which is unrealistic for the considered wait times [16, 35].

**Table 1. Model parameters and the value or range (sampled uniformly) used in this study (third columns).** Estimates from the literature with respective sample sizes (*n*) are reported where applicable and available (last column). * [33] considers only the U.S. adult population. Adding 0-17 year-olds with an assumed high-risk rate of 21.2% (the estimate for 18-59yr olds) and projected 2020 US census data [34] yields an overall high-risk estimate of around 1/3. † derived from Table 1 in [3] and projected 2020 US census data [34].

| Parameter | Meaning | Value/Range | References |
|---|---|---|---|
| **Interaction network parameters** | | | |
| N | population size | 10, 000 | |
| k | average number of private/public interactions | 4,6 (default),10 | [27] $2k \in [7.95, 19.77]$ |
| $p^{small-world}$ | probability of initial edge re-wiring in the private small-world network | 5% | |
| $p^{high\text{-}risk}$ | proportion high-risk individuals | 1/3* | [33] 37.6% (n = 430,000) |
| **Virus and disease parameters** | | | |
| $\beta_I$ | transmission rate of symptomatic individuals | [0.05, 0.4] | [16] 9.6% (n = 1,286) |
| $\beta_A$ | transmission rate of asymptomatic individuals | $[0, \beta_I]$ | [36] 46%-62% (of $\beta_I$) (n = 801) |
| $p_{E\to A}$ | proportion of asymptomatic infections | [5%, 50%] | [37] 20.6-39.9% (n = 634) |
| $\frac{p_{E\to A}^{low-risk}}{p_{E\to A}^{high-risk}}$ | ratio of asymptomatic infections in low-risk vs. high-risk individuals | [1, 5] | no data |
| $p_{I\to H}$ | probability of symptomatic individuals requiring hospitalization | 7% | [38] 5% (n = 44,415) [3, 39] 7.38%† |
| $\frac{p_{I\to H}^{high-risk}}{p_{I\to H}^{low-risk}}$ | ratio of high-risk vs. low-risk symptomatic individuals requiring hospitalization | [4, 10] | [3, 39] 6.47† |
| IFR | COVID-19 infection fatality rate (if hospitalized receive perfect care) | 1% | [39] 0.66% (n = 44,672) [3] 0.9% |
| $\frac{p_{H\to D}^{high-risk}}{p_{H\to D}^{low-risk}}$ | ratio of high-risk vs. low-risk hospitalized individuals dying from COVID-19 | [4, 10] | [3, 39] 6.33† |

## Virus transmission

Upon infection, individuals transition from one compartment to the next until they recover or die based on a stochastic process (Fig 1B). Susceptible individuals become infected through contact with contagious individuals and transition to the exposed compartment E.

**Table 2. Transition distributions.** Distributions used to model the time an individual spends in each transient, contagious compartment (all times in days). Mean (*μ*) and median (*m*) estimates and sample sizes (*n*) from the literature are reported where available.

| Parameter | Meaning | Distribution | Reported mean ($\mu$)/median ($m$) |
|---|---|---|---|
| $t_{E\to I}$ | time in exposed compartment if infection will be symptomatic | Poisson(5) | [15] $m = 4$ (n = 1,099) [16] $\mu = 5.95$, $m = 4.8$ (n = 183) [40] $m = 5.1$ (n = 181) [4] $\mu = 4.2$ (n = 140) [41] $\mu = 5.2$ (n = 49) |
| $t_{E\to A}$ | time in exposed compartment if infection will be asymptomatic | Poisson(5) | no data, assumed to be distributed as $t_{E\to I}$ |
| $t_{I\to H}$ | transition time from symptom onset to hospitalization | Poisson(8) | [16] $\mu = 4.64$, $m = 3.41$ (n = 391) [42] $m = 11$ (n = 191) [22] $m = 7$ (n = 138) [43] $m = 7$ (n = 41) |
| $t_{I\to R}$ | transition time from symptom onset to recovery (end of viral shedding) | Poisson(20) | [16] $m \in [17.5, 22.9]$ age-dependent (n = 228) [39] $\mu = 24.7$ (n = 165) [42] $m = 20$ (n = 137) |
| $t_{A\to R}$ | transition time from full viral shedding to recovery | Poisson(20) | no data, assumed to be distributed as $t_{I\to R}$ |
| $t_{H\to R}$ | transition time from hospitalization to recovery | Poisson(12) | [15] $\mu = 12.8$, $m = 12$ (n = 1,099) [4] $\mu = 11.5$ (n = 140) [42] $m = 12$ (n = 137) [44] $\mu = 17.4$ (n = 21) |

Contrary to SARS [45], recent reports indicate that SARS-CoV-2 can be transmitted before the onset of symptoms and by asymptomatic cases [37, 46, 47]. To account for early transmission potential in our model, individuals in compartments E, A, I and H may all transmit the virus, with transmission rates dependent on the time since infection. We assumed that exposed individuals (compartment E) become contagious 2 days before peak viral load, which coincides with symptom onset in symptomatic cases (i.e., the latent period is two days shorter than the incubation period). Transmission rates over time typically follow a Gamma distribution [48]. Based on preliminary data [35], we used a Gamma-distributed transmission rate with shape = 2 and scale = 2 (Fig 1c). We further assumed that asymptomatic cases cannot be more contagious than symptomatic ones. SARS-CoV-2 transmission rates, especially of asymptomatic cases, are currently not well understood [16], so we considered a range of values for the peak transmission rate of symptomatic cases (at symptom onset), $\beta_I \in U(0.05, 0.4)$ and a dependent range for asymptomatic ones, $\beta_A \in U(0, \beta_I)$ (that is, we picked $\beta_A/\beta_I \in U(0, 1)$).

Once exposed individuals reach peak infectivity they transition to the asymptomatic (A) or symptomatic (I) compartment. The proportion of asymptomatic COVID-19 infections is currently unknown; we therefore sampled the overall proportion of asymptomatic infections from a uniform distribution, $1 - p_{E\to I} = p_{E\to A} \sim U(0.05, 0.5)$ (parameterized based on available estimates from the literature; Table 1), and further sampled the ratio of asymptomatic infections in low-risk versus high-risk individuals from another uniform distribution, $p_{E\to A}^{\text{low-risk}}/p_{E\to A}^{\text{high-risk}} \sim U(1, 5)$.

## Hospitalization due to severe infection

A proportion of symptomatic individuals develop a severe infection requiring hospitalization; this rate may depend on the underlying health of the individual. We thus included two hospitalization parameters in the model (Table 1): $p_{I\to H} = 1 - p_{I\to R}$ describes the overall proportion of symptomatic cases that eventually develop a severe infection and require hospitalization, while $p_{I\to H}^{\text{high-risk}}/p_{I\to H}^{\text{low-risk}}$ describes the increased likelihood of a high-risk individual requiring hospitalization. We fixed $p_{I\to H} = 7\%$ and considered a range [4, 10] for the differential risk ratio [3, 38, 39].

## COVID-19-related mortality

We assumed that all COVID-19-related deaths occur after hospitalization–that is, individuals do not die before being hospitalized. This assumption is likely valid for most developed countries; however, this assumption could be violated in communities without easy access to hospital facilities.

We set the overall proportion of hospitalized individuals dying from COVID-19, $p_{H\to D}$, to align with a COVID-19 infection fatality rate (IFR) of 1% [3, 39]. That is,

$$p_{H\to D} = \frac{\text{IFR}}{p^{\text{high-risk}} p_{E\to I}^{\text{high-risk}} p_{I\to H}^{\text{high-risk}} + p^{\text{low-risk}} p_{E\to I}^{\text{low-risk}} p_{I\to H}^{\text{low-risk}}}.$$

This value for the IFR is likely an overestimate, but accurate fatality rates in an emerging epidemic are very difficult to estimate [49]. As before, we introduced a ratio describing the differential risk of dying from COVID-19 for high- and low-risk individuals, $p_{H\to D}^{\text{high-risk}}/p_{H\to D}^{\text{low-risk}} \sim U(4, 10)$ [3, 39]. To simulate deaths in our network model, we made the simplifying assumption that each day a severely infected person has the same chance of dying from COVID-19 (i.e., the time to death is geometrically distributed with the distribution

parameter corresponding to a per-day death rate). We set the per-day death rate for low-risk and high-risk individuals to align with $p_{H \to D}^{\text{low-risk}}$ and $p_{H \to D}^{\text{high-risk}}$, respectively.

## Model summary

Each day, the network model updates simultaneously as follows:

- Susceptibles may become infected through private or public, random interactions with contagious individuals. The interaction probabilities are based on the multi-layered interaction network.

- Newly infected individuals move to the exposed compartment (E) and risk-group-dependent random variables are drawn describing the future course and transition times of the infection.

- Hospitalized individuals die at a risk-group-dependent per-day death rate.

- The transition times to the next compartment of all contagious individuals are reduced by a day. Individuals with a transition time of zero transition to the next compartment.

To investigate the effects of social distancing, triaging, and testing, we added additional features to this base stochastic network model as follows.

## Social distancing

We modeled the general effects of social distancing policies with two parameters, private activity level $a^{private}$ and public activity level $a^{public}$, which describe the average degree to which an individual without symptoms (in compartments S, E, A or R) reduces private and public interactions in response to the pandemic. An individual who has not adopted social distancing behaviors has activity levels of 1, while perfect isolation corresponds to activity levels of 0.

We assumed that symptomatic individuals (in compartment I) further reduce their private and public activity levels due to symptoms and to avoid infecting others at an average rate of $r^{\text{symptoms}} \sim U(0, 1)$. Similarly, we assumed that severely infected individuals requiring hospitalization (in compartment H) are completely isolated. Finally, we assumed that individuals in the high-risk group may, independently of their compartment, choose to reduce their activity levels more than the low-risk group. We therefore included an additional high-risk activity reduction, $r^{\text{high-risk}} \sim U(0, 1)$. This variable also allows us to adjust for the generally reduced contact rates of older, more likely high-risk individuals [27].

The probability that two individuals who practice social distancing still meet, with the potential to infect one another, is given by a mass action-like product of their respective activity levels. For example, the probability that a symptomatic, low-risk individual meets a high-risk friend is given by $a^{private}(1 - r^{\text{symptoms}}) \cdot a^{private}(1 - r^{\text{high-risk}})$.

In reality, each person decides individually how to adapt their social behavior in response to COVID-19. For this reason we assigned activity levels to an individual (node) rather than a contact (edge). To compare policy efficacies, however, we combined all individual-based activity levels into an overall, population-wide contact reduction rate. In a population without symptomatic infections (i.e., at the start of the outbreak), this overall contact reduction rate is a function of the underlying interaction network, the private and public activity levels, the additional activity reduction of high-risk individuals and the proportion of high-risk individuals.

## Hospital triaging

The baseline model assumes unlimited healthcare resources, but in reality the number of hospital beds, ICU beds, ventilators, and trained health care professionals are limited. We assumed the healthcare system can provide perfect care if operating below a capacity threshold of 6 beds per 1000 individuals (3 beds for every 1000 individuals [50] and we assumed that this capacity threshold doubles during times of emergency).

Once the number of individuals requiring hospitalization (in compartment H) rises above the capacity threshold, overall care is sub-standard and triaging questions regarding resource allocation arise (Fig 1D). In the absence of data, we modeled the decrease in the average care provided per person with a square-root function, where average care per person $= 1/\sqrt{\text{hospital capacity}}$. We evaluated four triage options: (1) fill empty beds based on a wait list (first-come first-served), (2) fill empty beds randomly (lottery), (3) fill empty beds with least-severely infected based on clinical judgment (in the model, clinical judgement corresponds to the known remaining time to recovery), and (4) provide the same level of imperfect care to each individual (e.g., sharing of a single ventilator among multiple patients). Under the first three policies, patients receiving care will receive perfect care until they recover or die, and the sole difference among these three policies is how empty beds are allocated.

Hospitalized individuals (in compartment H) who received perfect care and who do not die on a given day move one day closer to recovery. Hospitalized individuals who received some level of care move a partial day closer to recovery, corresponding to the amount of imperfect care they received, while individuals who received only palliative care do not progress towards recovery at all.

## Testing

To evaluate the efficacy of testing policies in reducing COVID-19-related deaths, we assumed a fixed maximum number of tests available per day, and that testing begins as soon as the first person becomes symptomatic. We further assumed that, per CDC guidelines, severely infected individuals arriving at a hospital (compartment H) receive priority testing [12]. Remaining available tests are administered to symptomatic individuals (compartment I), and a shortage of tests precludes testing of individuals without symptoms (compartments S, E, A, and R). We compared two primary testing policies for symptomatic individuals: (i) test high-risk individuals first, or (ii) test low-risk individuals first. Further, to determine testing priority within each risk group, we compared two secondary testing policies: (i) test individuals in the order in which they developed symptoms (i.e., test first symptomatic first), or (ii) test individuals in the reverse order in which they developed symptoms (i.e., test recent symptomatic first). Finally, we included in the model a delay in test results of up to seven days.

While testing hospitalized individuals serves an essential clinical role, testing symptomatic individuals is solely preventive. Individuals who test positive are currently placed under quarantine, which in theory completely prevents virus transmission. In reality this is not always the case, especially when self-quarantine is conducted at home. We therefore included the average activity reduction of a positively tested individual as a further model parameter, and assumed a positive test yields a 80%–100% reduction in activity levels for the duration of the infection, in addition to the already-reduced activity levels due to symptoms, (i.e., $r^{\text{positive}} \sim U(0.8, 1)$). For example, the probability that a symptomatic, low-risk individual who tested positive meets a high-risk friend is given by $a^{private}(1 - r^{\text{symptoms}})(1 - r^{\text{positive}}) \cdot a^{private}(1 - r^{\text{high-risk}})$.

## Model analysis

Because the true values of many virus- and disease-related parameters are currently uncertain, we considered broad ranges for unknown parameters. To ensure sufficient coverage of the high-dimensional parameter space, we opted for a large number of sample points versus replication. The discrete and categorical parameters were sampled uniformly at random, while we used Latin Hypercube sampling [51] for the continuous parameters to ensure coverage by spreading the sample points across the parameter space (S1 Table). For private and public activity levels as well as the additional high-risk contact reduction, we applied a post-sampling transformation (S1 Table) to ensure the distribution of population-wide contact reduction, which is derived from these three parameters and the underlying network, was wide enough to be representative (S2I and S3I Figs).

For most analyses (Figs 2, 3, 4A, 5A and 5B), we ran the model $10^6$ times, each time with a different parameter setting. When comparing the effect of triage policies we balanced coverage with precision by sampling $10^3$ parameter settings and running the model 250 times for each parameter setting for each triage policy (S1 Table, Fig 4B). We initialized the runs for each triage policy with the same 250 random seeds to ensure identical disease dynamics up to the point where hospitals operate at overcapacity. To compare primary and secondary testing policies (Fig 5C), we followed the same approach, except that we sampled from a lower-dimensional parameter space. We fixed the maximum number of tests per day at 10, assumed no delay, and considered only 3 and 20 levels of additional contact reduction by symptomatic individuals and high-risk individuals, respectively (details in second-from-right column of S1 Table). Finally, when comparing the interaction between all policy domains (Fig 6), we sampled $10^3$ parameter settings following the same approach as before, and ran the model 100 times for each parameter setting and each of the 16 combinations of policy choices (0 vs. 40 maximal tests per day (2 choices); triage policies: imperfect but equal care vs. treat least severely infected first (2 choices); contact reduction levels: 0%, 25%, 50%, 75% (4 choices); details in the right column of S1 Table).

## Model outcomes

The primary model outcome considered in this study is the average number of deaths, or in relative terms the proportion of the population that dies from COVID-19. Related model outcomes, considered in Fig 4A, S1, S2, S3 and S4 Figs. include the proportion of the population infected with SARS-CoV-2, the COVID-19 infection fatality rate (%dead / % infected), the COVID-19 case fatality rate (%dead / %symptomatic), the peak hospital (over)capacity (peak %hospitalized / capacity threshold), the initial basic reproductive number (average number of secondary infections caused by the individual who initially imported SARS-CoV-2 into the population), the time at which half of all infections occurred (a measure of "flattening the curve" [2]), the average disease generation time (the time between infection of an individual and the time when the infecting person was infected), and the proportion of transmissions caused by asymptomatic cases (in compartment E or A).

## Quantitative analysis

The model was implemented and all model analyses were run entirely in Python 3.7. The contour plots in Figs 2, 3 and 4A were generated by binning the data using a 20x20 equidistant grid, and subsequent smoothing using a 2-dimensional Savitzky-Golay filter [52]. To avoid over-smoothing, we chose a small window size of 5 and used only linear functions. Similarly, we used a one-dimensional Savitzky-Golay filter with window size 200 and linear functions to

serve as a generalized moving average of the 1000 data points presented in Fig 4B. In Fig 5B and 5C, we summarized the raw data using a linear regression line.

## Supporting information

**S1 Fig. Average disease generation time versus various model parameters and outcome measures.** All model parameters were chosen as described in Tables 1 and 2 and in the third column of S1 Table. Bivariate Gaussian kernel density estimates are shown.
(PDF)

**S2 Fig. Histogram of select secondary model parameters (green) and outcome measures (blue).** All model parameters were chosen as described in Tables 1 and 2 and in the third column of S1 Table.
(PDF)

**S3 Fig. Histogram of select secondary model parameters (green) and outcome measures (blue) for non-contained epidemics.** All model parameters were chosen as described in Tables 1 and 2 and in the third column of S1 Table. Data were restricted to those model runs that resulted in at least 10% infected (63.7% of model runs).
(PDF)

**S4 Fig. Spearman correlation of various model parameters and outcome measures.** All model parameters were chosen as described in Tables 1 and 2 and in the third column of S1 Table.
(PDF)

**S1 Table. Description of the particular parameter space sampled for the different analyses.** Square brackets [] denote sampling from a continuous space, while curly brackets {} denote sampling from a discrete set of values. * Identical 1,000 parameter settings and identical 250 seeds were used to compare the different triage policies in Fig 4B, and the different combinations of primary and secondary testing policies in Fig 5C. In Fig 6, only 100 seeds were used to compare the different combinations of policy choices.
(PNG)

## Acknowledgments

We thank Rana Parshad and Zhijun Wu for fruitful initial discussions, Bernard Lidicky and Miles Aronnax for help with high performance computing, Katharina Kusejko, Thomas Liechti, Nancy Boury, Philip Dixon, and Carolyn Seyler for helpful conversations, comments, and clarifications.

The complete Python implementation of the model is available at Github at https://github.com/ckadelka/COVID19-network-model.

## Author Contributions

**Conceptualization:** Claus Kadelka.

**Formal analysis:** Audrey McCombs, Claus Kadelka.

**Investigation:** Audrey McCombs, Claus Kadelka.

**Methodology:** Audrey McCombs, Claus Kadelka.

**Software:** Claus Kadelka.

**Visualization:** Claus Kadelka.

**Writing – original draft:** Audrey McCombs, Claus Kadelka.

**Writing – review & editing:** Audrey McCombs, Claus Kadelka.

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
