## [Decision Letter · Decision Letter 0]

27 Aug 2020

Dear Dr. Kadelka,

Thank you very much for submitting your manuscript "A model-based evaluation of the efficacy of COVID-19 social distancing, testing and hospital triage policies" for consideration at PLOS Computational Biology. As with all papers reviewed by the journal, your manuscript was reviewed by members of the editorial board and by several independent reviewers. The reviewers appreciated the attention to an important topic. Based on the reviews, we are likely to accept this manuscript for publication, providing that you modify the manuscript according to the review recommendations.

In particular, please be sure to frame your work appropriately with regard to the existing literature.  I agree with the reviewers that there are other studies addressing multiple policy domains, and it seems that the novelty of your work falls more in investigating these in the context of a contact network model. 

In addition, please give a more robust discussion of the assumptions you have made in formulating this model, particularly with regard to network structure and dynamics, and the consequences for your findings.  At present you make some brief statements in the discussion and say this could be addressed in further work.  While true, this does nothing to advance our understanding of the results you present here; please add discussion of how these assumptions could have shaped your findings, and how we should interpret the robustness of your findings as a consequence. 

Finally, one reviewer states at several points that the model would be stronger if fit to real datasets.  I certainly agree, but see this as beyond the scope of your current theoretical study.  With that said, if you are able to add more case studies or points of contact to the empirical literature, it will increase the impact of your work. 

Sincerely,

James Lloyd-Smith

Associate Editor

PLOS Computational Biology

Nina Fefferman

Deputy Editor

PLOS Computational Biology

[LINK]

Reviewer's Responses to Questions

**Comments to the Authors:**

Reviewer #1: A model-based evaluation of the efficacy of COVID-19 social distancing, testing and hospital triage policies

Audrey McCombs, Claus Kadelka

General Comment: A timely model that would be very helpful as a framework for considering complex compartmental models that addresses multiple concerns at once. I would recommend it for publication after minor edits. Currently, in the US, it seems that all of the policies that have been implemented have worked against one another and the public in general seems confused as to what exactly is scientifically acceptable behavior during this difficult time. I think a model like this can help guide future policy implementation and keep the oversaturation of control measures from overwhelming the public.

Other comments:

Abstract: Line 1

Reword, phrase “SARS-CoV-2 and COVID-”. These are not two different diseases. One is the name of the virus and the other is the disease caused by the virus, see line 2.

Lines 9-15: May be lead this paragraph with what these policy domains are before saying no modeling efforts have looked at them simultaneously.

Line 51-52: It would be good to also explore how these policies impact case incidence

Line 53-55: I’m very interested to see how this model fairs with more accurate data and tailored to a specific city.

Line 74-75: consider adding an ie statement here to make things a bit more clear

Fig 2: Is it accurate to assume the same number of private and public contacts? This is discussed a bit later, but it would be nice to see the effects of having more/less public interactions for say, a politician, vs an average citizen in the US

Fig 3-4: Once everything is formatted, I think this section will flow very well

-It seems like some things could be moved to/from this section and the discussion section. Some explanation of the results is given, but not fully expanded on until the discussion section, which was a bit jarring, but did not retract from my understanding of the content in a major way

Line 167: I don’t see a Fig C anywhere. Is it fig 1, C? Or in the supplementary materials?

Line 187: seeing this model adapted to a particular school with better parameter estimates to see possible effects of school reopenings would be interesting

Lines 194: “We modeled the spread of an imported case”

This was not mentioned in the main text, not the effect this imported case

Line 195: have you tested your model with different sized populations to see if any of the dynamics change?

Line 196: closed and imported seem to be contradicting themselves here.

Lines 206-208: I think it’s important to include those individuals that properly quarantine and thus have close to 0 contacts per day, especially if a high-risk individual tests positive

Lines 212-214: What is the justification for the selection of these numbers?

Lines 219-220: See above, you simply initialized the model with one infected, the one infected is not due to importation

Line 222: See my comment above, not two different disease

Line 248-252: it would be interesting to explore the possibility that asymptotic individuals having a higher transmission rate

Line 237: Remove parenthesis on the statement (and transition to the exposed compartment E) them move it to line 238 after “contagious individuals”

Line 268-269: It is possible in less well-developed countries that infected individuals could die before being hospitalized, so this ratio may need to be adjusted to reflect that. It will be good to may a statement about this in the text.

Line 313: “her” - maybe use a gender neutral verb 'their'

Line 365-367: it may be that low risk individuals don’t have enough autonomy here. For example, even though I am a low risk individual, I would not meet with someone who tested positive (assuming that I find out, either through them telling me or social media, etc), so perhaps an extra term should be included to account for that

Line 367: Rewrite equation as written on line 312 before adding the (1 - rsymptoms)

Line 368: Model analysis

The detailed description of the model is good given all the moving parts However, the authors should write out the detailed model for clearer understanding of the model either in the main text or as a supplementary material.

Line 411: It will be helpful to readers if the authors can provide their source codes or a link to the source codes.

S2-3 Fig: The y axis could be relabeled to something a bit more informative, like disease incidence

S1 Table: Consider adding parameter descriptions with the table

-Condensing the explanations for the supplementary information in the pdf files would have been helpful

Reviewer #2: This is a very comprehensive study with some valuable insights. However, the amount of different policy domains touched on sometimes muddles some key points for that deserve more emphasis. For example, on pg. 3, consider the sentence "Furthermore, within each risk-group, testing recently-infected individuals first was more effective than prioritizing individuals who have been infected longer." This statement is fairly obvious and might not be a feasible policy choice. This sentence is either not needed or there should be more emphasis on the finding about superior results for public vs private contact reduction along with difference in benefits based on network structure. Overall this is a good paper that can be strengthened by emphasizing and expanding on key results, along with a few minor edits based on comments below.

Specific comments:

1. On pg. 2 line 10: Authors state "to our knowledge there are no studies examining

several policy domains simultaneously". This is a very naive statement given the amount of research conducted on COVID-19. One example, among several others: https://doi.org/10.1101/2020.06.10.20127860

2. pg. 3 line 60: "Our model yielded epidemiological outcome measures within the range of current 59

estimates [3, 4, 18]: an average initial basic reproductive number (R0) of 2.76 and an 60

average disease generation time of 5.29 days." The use of the word yielding is potentially misleading or unclear since R0 and generation time really depend directly on the input parameters.

**Have all data underlying the figures and results presented in the manuscript been provided?**

Reviewer #1: **No: **The authors used parameters from literature

Reviewer #2: None

PLOS authors have the option to publish the peer review history of their article (what does this mean?). If published, this will include your full peer review and any attached files.

Reviewer #1: No

Reviewer #2: **Yes: **Cameron Browne
---

## [Editor Report · Decision Letter 1]

27 Sep 2020

Dear Dr. Kadelka,

We are pleased to inform you that your manuscript 'A model-based evaluation of the efficacy of COVID-19 social distancing, testing and hospital triage policies' has been provisionally accepted for publication in PLOS Computational Biology.

Best regards,

James Lloyd-Smith

Associate Editor

PLOS Computational Biology

Nina Fefferman

Deputy Editor

PLOS Computational Biology

---

## [Editor Report · Acceptance letter]

12 Oct 2020

PCOMPBIOL-D-20-01075R1 

A model-based evaluation of the efficacy of COVID-19 social distancing, testing and hospital triage policies

Dear Dr Kadelka,

I am pleased to inform you that your manuscript has been formally accepted for publication in PLOS Computational Biology. Your manuscript is now with our production department and you will be notified of the publication date in due course.

With kind regards,

Matt Lyles
